

# A cloud-by-cloud approach for studying aerosol-cloud interaction in satellite observations

Fani Alexandri[1], Felix Müller[1], Goutam Choudhury[2], Peggy Achtert[1,3], Torsten Seelig[1], and Matthias Tesche[1]

[1]Leipzig Institute for Meteorology, Leipzig University, Stephanstraße 3, 04103 Leipzig, Germany
[2]now at Department of Geography and the Environment, Bar-Ilan University, Tel Aviv, Israel
[3]Meteorological Observatory Hohenpeißenberg, German Weather Service, Hohenpeißenberg, Germany

**Correspondence:** Matthias Tesche (matthias.tesche@uni-leipzig.de)

**Abstract.** The effective radiative forcing (ERF) due to aerosol-cloud interactions (ACI) and rapid adjustments (ERFaci) still causes the largest uncertainty in the assessment of climate change. It is understood only with medium confidence and studied primarily for warm clouds. Here, we present a novel cloud-by-cloud (C×C) approach for studying ACI in satellite observations that combines the concentration of cloud condensation nuclei ($n_{CCN}$) and ice nucleating particles ($n_{INP}$) from polar-orbiting lidar measurements with the development of the properties of individual clouds from tracking them in geostationary observations. We present a step-by-step description for obtaining matched aerosol-cloud cases. The application to satellite observations over Central Europe and Northern Africa during 2014 together with rigorous quality assurance leads to 399 liquid-only clouds and 95 ice-containing clouds that can be matched to surrounding $n_{CCN}$ and $n_{INP}$, respectively, at cloud level. We use this initial data set for assessing the impact of changes in cloud-relevant aerosol concentrations on the cloud droplet number concentration ($N_d$) and effective radius ($r_{eff}$) of liquid clouds and the phase of clouds in the regime of heterogeneous ice formation. We find a $\Delta \ln N_d / \Delta \ln n_{CCN}$ of 0.13 to 0.30 which is at the lower end of commonly inferred values of 0.3 to 0.8. The $\Delta \ln r_{eff} / \Delta \ln n_{CCN}$ between -0.09 and -0.21 suggests that $r_{eff}$ decreases by -0.81 to -3.78 nm per increase in $n_{CCN}$ of 1 cm$^{-3}$. We also find a tendency towards more cloud ice and more fully glaciated clouds with increasing $n_{INP}$ that cannot be explained by the increasingly lower cloud-top temperature of super-cooled liquid, mixed-phase, and fully glaciated clouds alone. Applied to a larger amount of observations, the C×C approach has the potential to enable the systematic investigation of warm and cold clouds. This marks a step change in the quantification of ERFaci from space.

## 1 Introduction

The optical and microphysical properties of clouds depend on the presence of atmospheric aerosol particles. Aerosols facilitate the formation of cloud droplets by acting as cloud condensation nuclei (CCN, Quinn et al. 2008). They also affect supercooled liquid and ice-containing clouds by acting as ice nucleating particles (INP, Kanji et al. 2017). Changes in the amount of aerosols that can act as CCN and INP through natural and anthropogenic emissions therefore influence the properties and the life cycle of clouds through aerosol-cloud interactions (ACI) and the accompanying rapid adjustments. The resulting effective radiative



forcing (ERF, Boucher et al. 2013) due to ACI (ERFaci) of -1.0±0.7 Wm$^{-2}$ is still assessed with only medium confidence and marks the largest uncertainty in our understanding of man-made changes to the Earth's energy budget (Forster et al., 2021).

ACI can be distinguished for warm (liquid-water) and cold (mixed-phase and ice) clouds. The ERFaci in warm clouds is defined as the disturbance in short- and long-wave radiation due to a change in cloud droplet number concentration ($N_d$) triggered by a change in aerosol concentration (Bellouin et al., 2020; Quaas et al., 2020). ERFaci consists of (i) the instantaneous radiative forcing (IRFaci) related to brightening of clouds in polluted environment (cloud albedo or Twomey effect, Twomey 1974) and (ii) the rapid adjustments of cloud fraction $f$, liquid water path $L$, and cloud top temperature $T_{\text{top}}$ (Albrecht, 1989;

Ackerman et al., 2004) in prompt reaction to the perturbation of the Twomey effect. The challenge lies in quantifying (i) the sign of the combined adjustments (Mülmenstädt and Feingold, 2018; Gryspeerdt et al., 2019) and (ii) the anthropogenic contribution to the disturbance in aerosol concentration.

So far, the Twomey effect has been studied most extensively. Other proposed ACI mechanisms suggest that smaller droplets in non-precipitating clouds can be transported to larger heights where they release latent heat for a stronger development of

convective clouds (cloud invigoration effect, Rosenfeld et al. 2008) or delay cloud glaciation (thermodynamic effect, Lohmann and Feichter 2005). At the same time, increased amounts of aerosols that can act as INP might lead to an earlier onset of cloud glaciation which increases the precipitation efficiency of the clouds (glaciation effect, Lohmann 2002). Depending on the type of aerosols and clouds, their interaction might either de- or increase the susceptibility for clouds to form rain (Rosenfeld et al., 2008). While observational evidence of the invigoration effect is still ambiguous (Fan et al., 2018; Douglas and L'Ecuyer,

2021; Igel and van den Heever, 2021; Romps et al., 2023), effects of aerosols on cloud glaciation have barely been studied on a global scale due to a lack of both suitable INP proxies (Villanueva et al., 2020) and reliable long-term observations of ice-containing clouds (Villanueva et al., 2021).

It is not only because of the conflictive impact of different adjustments that observation-based ACI studies are notoriously complicated. They require (i) separating between meteorological and aerosol effects on the observed clouds and (ii) establishing

causality of the identified processes (Koren et al., 2010). Different ACI mechanisms rarely occur in their pure form but rather act within a buffered system (Stevens and Feingold, 2009) which means that overlapping feedbacks might compensate or strengthen the overall effect. In addition, ACI effects are regime dependent (Stevens and Feingold, 2009), implying the need for a separated investigation of different cloud types and regimes.

There is a wealth of studies regarding the impact and relevance of different aerosol effects on clouds based on spaceborne

observations (Bellouin et al., 2020) with known challenges and clear visions for ways forward (Quaas et al., 2020; Rosenfeld et al., 2023). Spaceborne remote sensing is the only observational technique for gaining a global picture of ERFaci and provides the best available benchmark for evaluating the performance of global climate models (Quaas et al., 2009). The instruments in the polar-orbiting A-Train constellation of satellites form the work horses for most ACI studies (Rosenfeld et al., 2014, 2023; Bellouin et al., 2020; Quaas et al., 2020). These instruments provide observations of column-integrated parameters like aerosol

optical thickness (AOT), aerosol index (AI), cloud optical thickness (COT), cloud droplet number concentration $N_d$, cloud droplet effective radius $r_{\text{eff}}$, $L$, as well as insights into the vertical distribution of aerosol and cloud layers.



Spaceborne ERFaci studies employ sophisticated data processing to extract aerosol effects on clouds from noisy data and to establish causality of the identified processes (Quaas et al., 2020). In most cases, they investigate statistical relationships between cloud properties (i.e. albedo, COT, $N_d$, $r_{eff}$, and $L$) and aerosol parameters (AOT or AI) to resolve the underlying

mechanisms (Rosenfeld et al., 2014, 2023) – carefully accounting for effects of meteorological parameters (e.g., Koren et al. 2010; Yuan et al. 2011; Chen et al. 2014; McCoy et al. 2017).

We are far from reaching consensus in ERFaci even for warm clouds (Bellouin et al., 2020). As a result of the imperfect data from satellite observations (Quaas et al., 2020; Jia et al., 2021, 2022), there are major challenges that hamper progress in the satellite-based quantification of ERFaci.

First, there is a lack of properly quantifying those aerosols that interact with liquid clouds. Most studies of the Twomey effect rely on AOT or AI from passive remote sensing. However, these column-integrated parameters are unlikely to provide the CCN information needed for studying ACI (Shinozuka et al., 2015; Stier, 2016). Hence, height-resolved information on CCN concentrations is needed for further progress in understanding the effect of aerosols on warm clouds (Quaas et al., 2020). Such methodologies have been developed based on ground-based (Mamouri and Ansmann, 2016) and spaceborne lidar

measurements (Choudhury and Tesche, 2022a, b, 2023; Choudhury et al., 2022).

Second, ERFaci assessments generally exclude the effect of aerosols on cold clouds (Bellouin et al., 2020) due to a lack of reliable proxies for INP concentrations (Murray et al., 2021). However, a clear dependence of the fraction of glaciated clouds on the concentration of aerosols known to be efficient INP, such as mineral dust, is found from both ground-based (Kanitz et al., 2011) and spaceborne remote-sensing observations (Villanueva et al., 2020, 2021). This calls for a systematic investigation

of clouds in different regimes by using realistic metrics for the acting aerosols, i.e. INP concentrations instead of proxies. Such information can also be provided by lidar measurements (Mamouri and Ansmann, 2016; Marinou et al., 2019; Choudhury et al., 2022).

Third, observations of polar-orbiting instruments only provide snapshots and are incapable of resolving the temporal development of a cloud that is affected by a perturbation of the aerosol field. As a consequence, satellite-based studies of the

cloud-lifetime effect resort to the concept of precipitation susceptibility (L'Ecuyer et al., 2009) rather than the actual age of the cloud at the time of observation. The recent introduction of Lagrangian studies in which backward trajectories are used to account for the origin of observed cloud fields further highlights the need to consider temporal development for quantifying aerosol-cloud interactions (Christensen et al., 2020; Goren et al., 2019; Gryspeerdt et al., 2021, 2022). The tracking of clouds in geostationary satellite observations (Seelig et al., 2021, 2023) marks a way forward in more realistically considering cloud

development and lifetime in ACI studies.

The combination of polar-orbiting and geostationary observations resolves processes impossible to deduce solely from snapshots. It has been shown that aerosols from ship tracks and continental outflow cause polluted marine stratocumulus to form longer-lasting closed cells (Goren and Rosenfeld, 2012, 2015) and that the history of both the observed clouds and the air masses they are embedded in is vital for untangling ACI effects (Christensen et al., 2020). Observations with the

Spinning Enhanced Visible and InfraRed Imager (SEVIRI, Derrien and Gléau 2005) on Europe's geostationary Meteosat Second Generation (MSG) satellite have been employed to identify phase transitions in individually tracked convective clouds





(Coopman et al., 2019) and to identify at which $T_{\text{top}}$ glaciation is initiated (Coopman et al., 2020). Nevertheless, the potential of the time-resolved cloud observations with geostationary sensors has not yet been exploited sufficiently for studying ERFaci.

In general, three approaches are used for studying ACI from space. In the first one, spatially aggregated information on

aerosols and clouds – often for specific regions or cloud regimes – is correlated to identify relationships between the selected parameters (e.g., Quaas et al. 2004; Yuan et al. 2008; Costantino and Bréon 2010; Chen et al. 2014; Gryspeerdt et al. 2017) which are likely intrinsically biased (Jia et al., 2021; Goren et al., 2023; Gryspeerdt et al., 2023). The second one makes use of specific aerosol sources such as ships, industrial centres, or volcanoes during natural experiments (Christensen et al., 2022) but faces the challenge of how to expand the findings to the global scale. The third is focused on resolving hemispheric

differences under the premise that anthropogenic effects will only occur in the northern hemisphere (McCoy et al., 2020). Here, we are proposing a fourth approach – referred to as cloud-by-cloud (C×C) approach – in which data of (i) individually tracked clouds from geostationary satellite observations are matched with (ii) height-resolved and aerosol-type specific information on the concentration of CCN and INP around those clouds from polar-orbiting satellite observations (and (iii) fields from meteorological and aerosol reanalysis) to form a bottom-up data set of warm and cold clouds that can be stratified according to

different aerosol and cloud properties or meteorological parameters to investigate the Twomey effect, rapid adjustments, and aerosol effects on ice-containing clouds.

This paper starts with a description of the used data and methodologies as well as the general concept of employing the combination of geostationary and polar-orbiting satellite observations for ACI studies in Section 2. First results that demonstrate the new approach are presented in Section 3. The work finishes with a summary and an outlook in Section 4.

## 2   Data and methodology


ACI studies based on spaceborne observations conventionally aggregate temporally averaged (at least daily values) aerosol and cloud parameters (often from different sensors) within a coarse grid box (generally $1° \times 1°$) with meteorological parameters from reanalysis fields (Quaas et al., 2004, 2009; Koren et al., 2010; Rosenfeld et al., 2014, 2023; Bellouin et al., 2020) and investigate the change in cloud properties ($N_{\text{d}}$, $L$, $f$) with increasing aerosol concentration for situations with constrained

meteorological conditions (Gryspeerdt et al., 2016, 2017, 2019, 2021). In contrast, our C×C approach matches individual clouds (from geostationary observations) with their surrounding aerosol fields (from polar-orbiting observations) to form a bottom-up data set that allows for stratifying the data according to the parameters of interest for subsequent ACI studies.

Figure 1 gives an overview of the steps of our approach, provides essential references, and links towards to corresponding sections in the paper.

## 2.1   Cloud-tracking in geostationary satellite observations


The cloud-tracking toolbox of Seelig et al. (2021, 2023) is used to track individual clouds in time-resolved observations with geostationary sensors. The algorithm identifies objects in two consecutive time steps and uses Particle Image Velocimetry (Adrian and Westerweel, 2010) to create a velocity field between them in multiple overlapping windows of different sizes.



Matches between the objects from one time step to the next are created by considering the distance of the centroids of the
identified objects as well as their overlapping area. The best match within set thresholds is kept if the difference in area does
not exceed a factor of four. In case of a splitting event (one object is matched to multiple objects), one match is chosen
arbitrarily. In case of a merging event (multiple objects are matched to one object), the best match is kept. In case of a tie, again
one is chosen arbitrarily. This process is repeated for all consecutive time steps to form a trajectory of a tracked object.

The method has so far been applied to the binary cloud mask inferred from observations of MSG-SEVIRI (Seelig et al.,
2021) and the Advanced Baseline Imager (ABI, Schmit et al. 2017) aboard NASA's Geostationary Operational Environmental
Satellites – R Series (GOES-R-ABI) (Seelig et al., 2023). Cloud physical properties (CPP) can be derived from those observa-
tions only during daytime, as visible reflectances are needed for the retrieval. The CPP generally include $T_{\text{top}}$, cloud top height
($h_{\text{top}}$), cloud top pressure ($p_{\text{top}}$), COT, $r_{\text{eff}}$, and $L$. $N_d$ is derived using COT and $r_{\text{eff}}$ (Grosvenor et al., 2018; Quaas et al., 2020).
This work uses CPP derived from MSG-SEVIRI observations and provided in the CLAAS-2 (Benas et al., 2017) data set with
a temporal resolution of 15 minutes. These CPP are matched to the trajectories of tracked clouds to obtain mean and median
values per time step as well as for the lifetime of individual clouds.

For a first application of the C×C approach, we want to focus on low-level liquid clouds and mid-level mixed-phase clouds
as observed by MSG-SEVIRI over Europe, the Mediterranean, and Northern Africa. As outlined in Seelig et al. (2021, 2023),
the data set of all tracked clouds is filtered to obtain sub-samples that correspond to the targeted cloud types. First, a cloud
has to show a cloud top height that relates to the targeted cloud type. Second, a cloud has to form and dissolve in clear air to
assure that well-defined development can be observed. Finally, a cloud has to occur entirely during daytime so that CPP data
are available throughout its lifetime.

## 2.2 Cloud-relevant aerosol concentrations from spaceborne lidar observations

Ground-based lidar measurements provide great potential for inferring height-resolved CCN and INP concentrations (Mamouri
and Ansmann, 2016; Marinou et al., 2019). However, adaptation of the ground-based approach to spaceborne lidar observa-
tions is not straightforward as the required extinction-to-number-concentration conversion factors show strong regional vari-
ation (Ansmann et al., 2019). The Optical Modelling of CALIPSO Microphysics (OMCAM, Choudhury and Tesche 2022a)
algorithm provides an alternative approach for inferring CCN concentrations from spaceborne Cloud-Aerosol Lidar and In-
frared Pathfinder Satellite Observations (CALIPSO, Winker et al. 2009) lidar observations that is consistent within the wider
CALIPSO retrieval. Figure 2 presents a flow chart of the OMCAM algorithm. In short, quality-assured CALIPSO level
2 aerosol profiles are used to obtain aerosol-type specific extinction coefficients that constrain light-scattering calculations
(Gasteiger and Wiegner, 2018) based on the normalized size distribution and complex refractive indices for different aerosol
types in the CALIPSO aerosol model (Omar et al., 2009). The scaled size distribution relating to the modelled extinction coef-
ficient that best reproduced the measurement is then integrated from a lower size limit at which the different aerosol types are
likely to be relevant for cloud properties to obtain the number concentration of reservoir particles. The latter are then used as
input to commonly used parametrizations (DeMott et al., 2010, 2015; Steinke et al., 2015; Ullrich et al., 2017) to infer CCN



and INP number concentrations $n_{CCN}$ and $n_{INP}$, respectively. For more details on the OMCAM retrieval and its uncertainties, readers are referred to Choudhury and Tesche (2022a).

Thorough validation of OMCAM-derived $n_{CCN}$ generally finds values that are within a factor if 1.5 to 2.0 of independent airborne and ground-based in-situ observations (Aravindhavel et al., 2023; Choudhury et al., 2022; Choudhury and Tesche, 2022b), which is far less than their natural variability. OMCAM-derived data have been used to compile a global height-resolved $n_{CCN}$ climatology from 15 years of CALIPSO observations (Choudhury and Tesche, 2023) that can be used complementary to model-derived $n_{CCN}$ (Block et al., 2023) as a benchmark for cloud-resolved climate modelling.

For this work, OMCAM was expanded to give the concentration of INP reservoir particles and to subsequently use them in INP parametrizations to obtain $n_{INP}$ analogous to Mamouri and Ansmann (2016) and Marinou et al. (2019) (see Figure 2). In contrast to $n_{CCN}$, independent in-situ measurements of $n_{INP}$ are sparse. This is why the validation of OMCAM-derived values is still a matter of ongoing research. Nevertheless, initial findings indicate that CALIPSO-derived $n_{INP}$ have a quality comparable to those inferred from ground-based lidar measurements (Marinou et al., 2019; Choudhury et al., 2022).

## 2.3    Matching cloud trajectories with aerosol observations

The intercept points between individual cloud trajectories and the CALIPSO ground track are obtained with the TrackMatcher tool (Bräuer and Tesche 2022, https://github.com/LIM-AeroCloud/TrackMatcher.jl.git). The purpose of TrackMatcher is the identification of intercept points between two lines on a latitude-longitude grid and the collocation of the respective data fields along those tracks. The algorithm allows for finding intersections in any pair of tracks that provide information on time ($t$), latitude ($\varphi$), longitude ($\lambda$), and height ($h$). The main steps of the algorithm are to (i) load track data related to two platforms, (ii) interpolate the individual tracks using a piecewise cubic Hermite interpolating polynomial, (iii) find intersections by minimising the norm between the different track point coordinate pairs, and (iv) extracting auxiliary information at or around the intercept point as set by the operator. In addition, the accepted time difference between the two tracks can be chosen to best suit the application. In this study, we didn't limit the time difference for finding matches between a cloud track and the CALIPSO satellite but most cases fell within a range between 0 and ±90 minutes.

An example for matching a cloud trajectory with a CALIPSO measurement is presented in Figure 3. CALIPSO level 2 data are available in 5-km intervals along the ground track while the example trajectory consists of four time steps. When a match is made, CALIPSO profiles are filtered for quality-assured data at the height levels of the paired cloud. Aerosol data might be absent or dismissed in case of low aerosol load (low signal-to-noise ratio), retrievals with low confidence score, or cloud presence. In Figure 3, the cloud changes from purely liquid (time steps 1 and 2), over mixed-phase (time step 3), to purely ice (time step 4). Aerosol information is available in the vicinity of the matching point and can be used to infer mean CCN or INP concentrations surrounding the tracked cloud. Further details on this procedure are provided below.

## 2.4    Quality-assured development of matched clouds

Our cloud tracking (Section 2.1) is generally performed on a binary cloud mask in which pixels are differentiated as either cloud or cloud-free. In passive observations, cloudy pixels are not unambiguously referring to a specific cloud type (Rossow





and Schiffer, 1999). Instead, the column information observed at the top of the atmosphere might actually be the product of contributions of very different clouds, such as low-level water clouds and high-level ice clouds. While these contributions will not be discernible in a cloud mask, they will propagate into the inferred cloud physical properties. Clouds that are found to intercept a CALIPSO overpass (Section 2.3) therefore have to undergo quality assurance before they can be considered for further analysis.

This quality-screening is performed in two steps. Clouds are initially sorted according to their phase and subsequently investigated for realistic development of their physical properties. The *phase categorization* sorts clouds as:

1. **liquid-water** if (i) the phase of all pixels is liquid or (ii) the median cloud top temperature is warmer than -5°C

2. **mixed-phase** if there are both liquid and ice pixels in the temperature range from -5°C to -38°C

3. **ice (heterogeneously frozen)** if there are only ice pixels in the temperature range from -5°C to -38°C

4. **ice (homogeneously frozen)** if the median cloud top temperature of all pixels is lower than -38°C

The objective of our C×C approach is to investigate liquid-water clouds for their response to changes in CCN concentrations and mixed-phase and heterogeneously frozen ice clouds for their response to changes on INP concentrations. To obtain meaningful results, however, we need to exclude those clouds that appear to evolve non-physically, i.e. that show a large variation in their properties from time step to time step or throughout their lifetime. We have therefore identified four objective criteria

for assessing *realistic cloud development* based on cloud-top temperature, area, and phase. A cloud is flagged, if it shows

1. **unrealistic development of $T_{\text{top}}$ from start to end**: the norm of the difference between median $T_{\text{top}}$ at cloud beginning and end exceeds 30 K

2. **unrealistic development of $T_{\text{top}}$ between time steps**: the norm of the change in median $T_{\text{top}}$ from time step to time step exceeds 15 K

3. **unrealistic spread of $T_{\text{top}}$ within individual time steps**: the spread in $T_{\text{top}}$ (maximum $T_{\text{top}}$ - minimum $T_{\text{top}}$ per time step) exceeds 35 K for more than 50% of all time steps

4. **unrealistic development of cloud size**: the change in cloud size from time step to time step for clouds that consist of more than four pixels exceeds 100%, i.e. cloud size should neither double nor half

These criteria apply to the entire trajectory. A cloud is kept for by-eye inspection if one of criteria 1 to 4 is met. It is dismissed

(considered as unrealistic), if more than one of the criteria is met. The latter also applies if criterion 2 or 4 are met multiple times.

Figure 4 presents the development of two tracked clouds to illustrate the quality-assurance methodology. The example cloud on 14 August 2014 shows a development that we consider as realistic. The cloud was tracked from its formation at 1315 UTC until it dissolved at 1645 UTC. Median, minimum, and maximum $T_{\text{top}}$ all vary within a reasonable range throughout all 15-min

time steps. The cloud is partly or fully glaciated at temperatures where this is not unlikely. The change in cloud size is indicative





of plausible growth and decay rather than merging and splitting. Overall, this case represents a heterogeneously frozen ice cloud that can be used in studies of the effect of surrounding INP concentrations on cloud properties and development. In contrast, the example cloud on 24 March 2014 raises some of the flags listed above. Over the length of its existence from 0845 UTC to 1130 UTC, median $T_{\text{top}}$ shows multiple time-step-by-time-step changes of more than 15 K (criterion 3). This indicates the

coinciding presence of low and high clouds within the cloud area or even on pixel basis and would already lead us to dismiss that cloud as unrealistic. The spread in $T_{\text{top}}$ per time step exceeds 35 K for 10 out of 12 time steps (criterion 4) provides another indicator that clouds at different height levels are contributing to the column signal detected in the passive observation. Finally, the areal growth from time step 2 to time step 3 exceeds a doubling (criterion 2) and the norm of the difference in $T_{\text{top}}$ at the first and last time steps is very close to 30 K (criterion 1).

The consideration for realistic cloud development puts another constraint on the selection of cases that can actually be used in the formation of our bottom-up data base. Overall, the initial matching of trajectories to CALIPSO overpasses, followed by the demand for valid aerosol observations at cloud level, and the subsequent requirement for realistic cloud development reduce the data base from an order of a million tracked clouds to an order of a few hundred cases for consideration in our C×C approach. We have opted for the exceptionally high level of scrutiny despite the resulting poor data yield to assure that findings

that are later inferred from the bottom-up data base are physically meaningful. In future, the length of the available time series of around 15 years, the possibility to extend the CCN and INP retrievals to newer spaceborne lidar missions, the option for expanding the study area, and application to further geostationary sensors provide ample opportunity for extending the number of cases without compromising their quality.

## 2.5 Study region

For testing the new C×C approach, we have selected a region that covers Europe and northern Africa. Figure 5 shows that our study region extends from the Atlantic coast to eastern Ukraine and from southern Sweden to the southern tip of Tunisia and the Sinai peninsula. All four cloud categories listed in Section 2.4 are found in the study region. In addition, the region covers the pathways of mineral dust transport to Europe. This is of particular interest to our work as mineral dust particles are very efficient INP and most likely to cause detectable changes in cloud phase.

## 2.6 Deriving matched aerosol and cloud parameters

The data obtained from matching the information related to tracked cloud properties and cloud-relevant aerosol properties are presented based on the case study of 18 March 2014 shown in Figure 6. A liquid-water cloud could be tracked for four time steps between 1100 and 1145 UTC. Its first time step shows a median $T_{\text{top}}$ of about 14 °C which corresponds to a $h_{\text{top}}$ of about 500 m. The median $r_{\text{eff}}$ for this initial time step is 12.5 µm. Over the next three time steps, $T_{\text{top}}$ decreases to 2 °C as

$h_{\text{top}}$ increases to 2.2 km. At the same time, $r_{\text{eff}}$ decreases slightly to 12 µm. Throughout its lifetime, the cloud consisted of 18 pixels of which none revealed the presence of ice. The CALIPSO overpass matched with this trajectory occurred at 1137 UTC between the third and fourth time step of cloud lifetime. Around the overpass, CALIOP measurements reveal the presence of aerosols and clouds with upper-level clouds attenuating the lidar signal before it can reach the surface (Figure 6b). The





OMCAM retrieval gives an estimated $n_{CCN}$ of around $2000\,cm^{-3}$ between the surface and $2.5\,km$ height – well within the
height range of the observed cloud.

Pixel-wise cloud properties are matched to the aerosol field within an along-track distance of 13 5-km CALIPSO intervals
and 5 60-m height bins around the pixel's $h_{top}$. Valid data in CCN concentration within such a box is averaged to obtain a
median $n_{CCN}$ that can be related to the cloud properties. In the example in Figure 6, 14 cloud pixels show values of $h_{top}$ that
allow for assigning a median $n_{CCN}$ to their cloud properties while 4 pixels revealed values of $h_{top}$ that fell within a height range
for which no CCN information could be retrieved. Consequently, the case of 18 March 2014 contributes 14 pairs of matched
aerosol-cloud information to the statistical analysis presented below.

To summarize, the C×C approach allows for decomposing clouds at three stages:

1. the **pixel stage** considers individual cloud pixels and the 13-profiles-by-5-height-bins CALIPSO aerosol around the
   pixel's $h_{top}$. This is referred to as pixel-wise information in this paper.

2. the **time-step stage** considers cloud and aerosol mean and median values for individual time steps of a considered cloud.
   This information is not used in this paper.

3. the **trajectory stage** considers cloud and aerosol mean and median values for the entirety of a cloud trajectory, i.e., all
   time steps. This is referred to as whole-trajectory information in this paper.

## 3 Results

### 3.1 General overview

For a first application of the C×C approach, we want to focus on (i) liquid clouds which have been extensively studied with
other methods and (ii) ice-containing clouds for which our new approach will enable assessing the impact of INP concentra-
tions on cloud-phase and development. The tracking of clouds in CLAAS-2 data (Section 2.1) within the region of interest
(Figure 5) for 2014 gives a total of 8,924,639 trajectories of clouds with well-defined start and end, i.e. forming and dissolving
in clear air. The matching procedure outlined in Sections 2.3 implies rigorous screening of that data set to trajectories that
coincide with a CALIPSO observation during both day and night. This constraint reduces the amount of data to 4393 tracked
and matched clouds that could potentially be used for further investigation. Out of those, 964 daytime trajectories of liquid
clouds are found to be vertically co-located with CCN layers (Section 2.2). Further quality-assurance regarding realistic cloud
development (Section 2.4) finally gives 399 trajectories for further analysis. In the case of ice-containing clouds, 844 daytime
trajectories could be matched to vertically co-located INP layers of which 95 clouds remained after quality assurance (Sec-
tion 2.4). Following the phase categorisation in Section 2.4, the ice-containing clouds can be further separated into 42 clouds
that are homogeneously frozen, 8 clouds that are heterogeneously frozen, and 45 mixed-phase clouds. The data set of liquid
clouds also contains 37 trajectories of supercooled clouds for which neither the median $T_{top}$ per time step exceeds $0\,°C$ nor
cloud ice is detected. These supercooled liquid clouds provide a reference in studying the impact of $n_{INP}$ on the ice-containing
clouds.



Figure 5 gives an overview of the cloud trajectories that could be matched to aerosol information from CALIPSO measurements over Europe and north Africa for the year 2014. We will start with a description of the matched data sets before presenting first results of applying the C×C approach for studying liquid-water and ice-containing clouds.

## 3.2 The matched aerosol-cloud data sets

The data set of matched cases for the year 2014 within the study region in Figure 5 consists of 399 trajectories of liquid clouds and 95 trajectories of ice-containing clouds. Both cloud types are rather small and short-lived. They span lifetimes from 3 to 47 (3 to 20 steps for ice-containing clouds) time steps of 15 min with a median of 4 time steps. The histogram of trajectory length in Figure 7 shows that the majority of clouds exists for less than 2 h. The size of the liquid clouds per time step ranges from 1 to 341 pixels with a median of 5 pixels (not shown). Clouds consisting of 10 pixels or less make up 70% of the data set. The ice-containing clouds considered here are smaller than the liquid clouds with sizes ranging from 1 to 53 pixels, a median of 4 pixels, and 90% of clouds consisting of 15 pixels or less (not shown).

Figure 8 gives an overview of the distribution of $h_{\text{top}}$ and $T_{\text{top}}$ of the liquid and ice-containing clouds that could be matched with information of the surrounding aerosol field. Most of liquid clouds persist below 1.0 km. The median $h_{\text{top}}$ is 0.6 km while the occurrence of few pixels with large $h_{\text{top}}$ increases the mean to $0.9 \pm 1.1$ km. The majority of cloud pixels (95%) show positive values of $T_{\text{top}}$ with a mean of $21.0 \pm 12.4\,^{\circ}$C and a median of $20.9\,^{\circ}$C. These comparably high temperatures are related to the fact that a large number of tracked clouds is located over northern Africa (see Figure 5). Cloud pixels with negative $T_{\text{top}}$ in the liquid data set refer to super-cooled clouds which can be used as reference in the investigation of the effect of $n_{\text{INP}}$ on ice-containing clouds. When data are aggregated to represent the median values of entire trajectories, $h_{\text{top}}$ slightly increases to $1.3 \pm 1.5$ km with a median of 0.8 km while $T_{\text{top}}$ decreases correspondingly to $17.4 \pm 15.0\,^{\circ}$C with a median of $16.9\,^{\circ}$C. The overview of the ice-containing clouds in Figure 8b shows that those clouds generally occur at larger heights and lower temperature than the liquid clouds. One group of ice-containing clouds occurs at $T_{\text{top}} < -38\,^{\circ}$C and $h_{\text{top}} > 8$ km – in line with the regime of homogeneous freezing. The other group with $T_{\text{top}} > -38\,^{\circ}$C covers the regime of heterogeneous freezing and represents the temperature range at which mixed-phase and ice clouds can be observed. Those clouds make up 2673 pixels (53 clouds) with $h_{\text{top}}$ ranging from 0.4 to 9.7 km (0.8 to 9.2 km) and a median of 5.8 km (5.7 km). When considering all ice-containing clouds, the pixel-wise (whole-trajectory) mean values are $h_{\text{top}} = 6.8 \pm 2.4$ km ($7.5 \pm 2.5$ km) and $T_{\text{top}} = -23.6 \pm 18.5\,^{\circ}$C ($-30.0 \pm 18.8\,^{\circ}$C) with median value of 6.6 km (7.2 km) and -17.2$\,^{\circ}$C (-25.2$\,^{\circ}$C).

The pixel-wise and whole-trajectory histograms of COT and $r_{\text{eff}}$ for both liquid and ice-containing clouds are presented in Figure 9. Because retrievals of cloud microphysical properties require data at visible wavelengths, this information is available only during daytime. Hence, and as a result of measurement restrictions such as weak signal-to-noise ratio, not all tracked pixels provide meaningful data. The majority of liquid clouds is optically thin with a median of 0.92 and 0.74 and means of $2.38 \pm 5.23$ and $2.47 \pm 3.83$ for the pixel- and trajectory-based analysis, respectively. About 75% of all cases are below the mean value. For ice-containing clouds, these values increase to pixel- and trajectory based medians of 4.88 and 1.85, respectively, and means of $8.26 \pm 10.77$ and $3.94 \pm 4.86$. An even larger difference is revealed in the size of the cloud particles. For liquid clouds, the median $r_{\text{eff}}$ is 7.8 $\mu$m for both the pixel- and trajectory-based analysis with means of $10.1 \pm 5.0\,\mu$m and $9.8 \pm 4.0\,\mu$m,



respectively. Only 15% of cases show median values larger than $12\,\mu$m which indicates the likely presence of precipitation
(Rosenfeld et al., 2012). The presence of ice in clouds increases the inferred values of $r_{\mathrm{eff}}$ to pixel- and trajectory-based means
of $20.9\pm7.5\,\mu$m and $22.0\pm5.7\,\mu$m and medians of $23.2\,\mu$m and $24.4\,\mu$m, respectively.

The histogram of $n_{\mathrm{CCN}}$ for the 399 matched liquid clouds in Figure 10 shows that the C×C approach is capable of covering
a wide range of CCN conditions. While values range from 17 to $25721\,\mathrm{cm}^{-3}$ with a median of $872\,\mathrm{cm}^{-3}$, the majority of cases
(about 90%) are in a realistic range below $3000\,\mathrm{cm}^{-3}$. In the analysis of our data set, we will use $n_{\mathrm{CCN}}$ to discriminate between
clean and polluted conditions. This is done in two steps. First, the upper and lower 5% of cases are dismissed from the data set as
they are likely to represent unrealistic values that result from propagating unreliable measurements through the analysis chain.
For the present data set, this means that $n_{\mathrm{CCN}} < 154\,\mathrm{cm}^{-3}$ and $n_{\mathrm{CCN}} > 4409\,\mathrm{cm}^{-3}$ and the related cloud data are omitted from
further analysis. Second, the remaining 90% of cases are split into three ranges that represent clean (lower quantile), moderate
(middle quantile), and polluted conditions (upper quantile). The resulting ranges contain about 4800 matched data points each
and are $154\,\mathrm{cm}^{-3} \leq n_{\mathrm{CCN}} < 610\,\mathrm{cm}^{-3}$, $610\,\mathrm{cm}^{-3} \leq n_{\mathrm{CCN}} < 1210\,\mathrm{cm}^{-3}$, and $1210\,\mathrm{cm}^{-3} \leq n_{\mathrm{CCN}} < 4409\,\mathrm{cm}^{-3}$.

Figure 11 presents the histogram of $n_{\mathrm{INP}}$ related to the 95 ice-containing clouds. The majority of clouds (81%) occurs in
aerosol fields that relate to values of $n_{\mathrm{INP}}$ between $10^{-3}$ and $10\,\mathrm{L}^{-1}$ which are realistically found at atmospheric conditions
(Kanji et al., 2017). About 3% of inferred values fall outside the presented range of seven orders of magnitude (2% lower,
1% higher). These values are likely the result of unreliable CALIPSO retrievals or originate from extrapolating the applied
INP parametrizations to temperatures at which they are no longer applicable (Marinou et al., 2019). As in the case of $n_{\mathrm{CCN}}$,
unrealistic $n_{\mathrm{INP}}$ are excluded by dismissing the lowest and highest 5% of values and related cloud data from further data analy-
sis. As mentioned earlier, the detailed evaluation of the number concentration of reservoir particles and $n_{\mathrm{INP}}$ with independent
in-situ measurements analogous to Aravindhavel et al. (2023), Choudhury et al. (2022), and Choudhury and Tesche (2022b) is
ongoing and will be the focus of a future publication.

### 3.3  CCN effects on liquid-water clouds

Now that we have obtained a data set of matched cloud and aerosol observations, we can start to investigate cloud properties
for different ranges of $n_{\mathrm{CCN}}$. Figure 12 shows that the mean $n_{\mathrm{CCN}}$ in the three quantiles mentioned before are $403\pm110\,\mathrm{cm}^{-3}$,
$875\pm189\,\mathrm{cm}^{-3}$, and $2123\pm733\,\mathrm{cm}^{-3}$. If the values of $r_{\mathrm{eff}}$ that have been matched to the aerosol observation are grouped
according to those three $n_{\mathrm{CCN}}$ ranges, we can see a clear decrease in $r_{\mathrm{eff}}$ and an increase in $N_{\mathrm{d}}$ from clean to polluted aerosol
conditions that is in line with the Twomey effect. While $r_{\mathrm{eff}} = 11.78 \pm 6.6\,\mu$m (median of $9.0\,\mu$m) and $N_{\mathrm{d}} = 68 \pm 166\,\mathrm{cm}^{-3}$
(median of $42\,\mathrm{cm}^{-3}$) for the lower third of $n_{\mathrm{CCN}}$ observations, values decrease to $9.99\pm4.72\,\mu$m (median of $8.0\,\mu$m) and $75 \pm$
$100\,\mathrm{cm}^{-3}$ (median of $53\,\mathrm{cm}^{-3}$) for the middle third, and even further to $9.34\pm3.86\,\mu$m (median of $7.8\,\mu$m) and $95 \pm 199\,\mathrm{cm}^{-3}$
(median of $60\,\mathrm{cm}^{-3}$) for the upper third (see Figure 12).

This allows for a first quantification of the change in $r_{\mathrm{eff}}$ and $N_{\mathrm{d}}$ related to a change in $n_{\mathrm{CCN}}$ rather than aerosol proxies such
as AOT or AI generally used in ACI studies based on satellite data (Bellouin et al., 2020; Quaas et al., 2020). An overview
of the inferred sensitivities of $r_{\mathrm{eff}}$ and $N_{\mathrm{d}}$ to changes in $n_{\mathrm{CCN}}$ is provided in Table 1. We find a $\Delta r_{\mathrm{eff}}/\Delta n_{\mathrm{CCN}}$ in the range
from -0.81 to -3.78 $\mathrm{nm}\,\mathrm{cm}^{-3}$ depending on the considered quantiles and the use of median or mean values of the distributions





in Figure 12. In the same way, $N_d$ is found to increase between 1.17 and 2.30 per increase of $100\,\mathrm{cm}^{-3}$ in $n_{\mathrm{CCN}}$. The larger
sensitivities found when derived over just the lower two quantiles (columns 4 and 6 in Table 1) corroborate that $r_{\mathrm{eff}}$ and $N_d$ are
most susceptible to changes in CCN concentration for low baseline values and that the effect becomes saturated with further
increasing $n_{\mathrm{CCN}}$ (Gryspeerdt et al., 2023).

For better comparison to studies based on passive CCN proxies, Table 1 also provides sensitivities for logarithmic changes.
The analysis of the C×C data set gives values of $\Delta \ln r_{\mathrm{eff}}/\Delta \ln n_{\mathrm{CCN}}$ between -0.09 and -0.21 while $\Delta \ln N_d/\Delta \ln n_{\mathrm{CCN}}$ is
found to vary in the range from 0.13 to 0.30. The latter values are at the lower end of earlier findings of 0.3 to 0.8 based on
column CCN proxies (Bellouin et al., 2020). However, they are the first ones to use actual CCN concentrations at cloud level
and the isolated clouds considered in the current data set might not be as sensitive to changes in aerosol concentration as, e.g.,
widespread stratocumulus clouds (Bellouin et al., 2020; Gryspeerdt et al., 2017; Goren et al., 2019). Future research based on
a larger C×C data set will reveal if the lower sensitivity can be corroborated and if the regional variation of sensitivity of cloud
parameters to changes in $n_{\mathrm{CCN}}$ can also be resolved with our new approach.

Applying the same analysis to COT (mean/median of around 2/1), cloud lifetime (mean/median of 8/5 time steps), and cloud
top height (mean/median of 800 m/600 m) does not reveal any change of those parameters with changing aerosol concentration
(not shown). For $L$, we find a decrease of the mean from $25\pm47\,\mathrm{g\,m}^{-2}$ for clean conditions to $16\pm37\,\mathrm{g/m}^2$ for polluted
conditions. However, median values show comparable values around $5\,\mathrm{g\,m}^{-2}$ for all three intervals of aerosol concentration in
line with optically thin clouds. While these findings are not in line with the assumption that increased $n_{\mathrm{CCN}}$ increases cloud
lifetime, albedo (COT), $h_{\mathrm{top}}$, and $L$, we consider the results as still preliminary. The C×C approach will need to be applied to a
longer time series and over a larger region to extend the data set for ACI studies from which more robust conclusions regarding
the adjustments can be drawn. This will be the focus of follow-up work.

The use of CALIPSO observations furthermore allows for allocating $n_{\mathrm{CCN}}$ to a specific aerosol type. In the data set presented
here, mineral dust and polluted continental make up the most of the CCN population. Therefore, an aerosol-type specific
investigation of the connection between $n_{\mathrm{CCN}}$ and $r_{\mathrm{eff}}$ will also be left for follow-up studies with a larger C×C data set.

### 3.4 INP effects on ice-containing clouds

The capability of the C×C data set to provide information on $n_{\mathrm{INP}}$ and the development of the phase of a tracked cloud (Section 2.4) opens new avenues for studying the effect of aerosols also on ice-containing clouds. However, the related constraints
for quality-assured cloud development and the lower abundance of successful INP retrievals from CALIPSO observations lead
to a data set that is much smaller than the one for studying effects of changes in $n_{\mathrm{CCN}}$ on the properties of warm liquid-water
clouds. Keeping this in mind, the current C×C data set for one year of data can only provide an outlook of its potential for
spaceborne ACI studies.

The impact of $n_{\mathrm{INP}}$ on different cloud types in the regime of heterogeneous ice nucleation is illustrated in Figure 13. We
consider all clouds in Figure 8 for which the whole-trajectory $T_{\mathrm{top}}$ as well as $T_{\mathrm{top}}$ per time step never exceeded 0°C and for
which all pixels are classified as liquid as super-cooled. These conditions are fulfilled by 42 clouds. Mixed-phase clouds have
to show a combination of pixels that are classified as liquid and ice (see Figure 3 for an example). This is the case for 45



clouds. Heterogeneously frozen clouds feature only pixels that are classified as ice and have to show values of $T_{\text{top}}$ larger than -38°C, which is the case for 8 clouds. Super-cooled clouds show a mean (median) $T_{\text{top}}$ of $-11.7 \pm 15.7$°C ($-6.2$°C). Mixed-

phase clouds are colder with $T_{\text{top}}$ of $-14.0 \pm 7.5$°C ($-11.2$°C) while heterogeneously frozen clouds show the lowest $T_{\text{top}}$ of $-17.1 \pm 3.7$°C ($-16.9$°C). These clouds occur in an aerosol environment with $n_{\text{INP}}$ of $0.04 \pm 0.11\,\text{L}^{-1}$ (median of $0.02\,\text{L}^{-1}$), $0.23 \pm 0.31\,\text{L}^{-1}$ ($0.05\,\text{L}^{-1}$), and $2.64 \pm 1.79\,\text{L}^{-1}$ ($2.92\,\text{L}^{-1}$), respectively.

The parametrizations used to infer $n_{\text{INP}}$ from the CALIPSO observations are strongly dependent on temperature. For mineral dust, they generally show an increase in $n_{\text{INP}}$ by about 1 order of magnitude per 5°C decrease (Marinou et al., 2019). The

INP data set presented here consists primarily of CALIPSO observations in which the detected aerosols are classified as mineral dust. Hence, we would expect a similar relation between the inferred $T_{\text{top}}$ and $n_{\text{INP}}$ of different cloud types if there was no change in the amount of reservoir particles that could become INP-active, i.e., if there was no increase in aerosol concentration. Instead, we find that mean $n_{\text{INP}}$ changes by about 1 order of magnitude already for a 3°C decrease in mean $T_{\text{top}}$. This indicates that the increase in $n_{\text{INP}}$ observed in the presence of mixed-phase clouds compared to super-cooled clouds and

of heterogeneously frozen clouds compared to mixed-phase clouds is not purely related to the decreasing $T_{\text{top}}$ of the different cloud types but also to an increase in the concentration of aerosols that can act as INP. In return, we can reason that the increased aerosol concentration facilitates earlier cloud glaciation as clouds surrounded by the largest abundance of INP are fully glaciated while those with medium and low $n_{\text{INP}}$ are of mixed-phase and liquid phase, respectively. This is in agreement with studies in which dust concentrations from aerosol reanalysis fields is used as a proxy for the presence of INP (Seifert et

al., 2010; Villanueva et al., 2020).

As in the case of liquid clouds, a larger data set is needed to obtain more robust constraints on the effect of aerosol concentrations on the properties of ice-containing clouds. This would facilitate, e.g., to assess if variation in $n_{\text{INP}}$ can also be found for super-cooled, mixed-phase, and heterogeneously frozen clouds with comparable $T_{\text{top}}$, which would be a more direct indication of the effect of aerosol concentration on cloud phase.

## 410   4   Summary and outlook

We present a detailed description of a new approach for spaceborne ACI studies in which cloud development from geostationary satellite observations is matched with height-resolved CCN and INP concentrations from polar-orbiting satellite observations on a cloud-by-cloud (C×C) basis. The novel C×C approach has been applied to observations of MSG-SEVIRI and the CALIPSO lidar over Europe and Northern Africa for the year 2014. Apart from the spatio-temporal matching of the

observations, the analysis chain includes further screening to assure that the resulting data set contains only (i) realistically developing clouds during daytime for which macro- and microphysical properties can be inferred throughout their lifetime and (ii) scenarios in which cloud-relevant aerosols actually occur within the height range of the matched cloud.

Tracking objects in the CLAAS-2 cloud mask gives about 9 million clouds with well-defined start and end, i.e. forming and dissolving in clear air, over the considered region throughout one year. Further data processing steps outlined in this paper

reduce this number to 399 liquid clouds and 95 ice-containing clouds that can be matched to surrounding concentrations of





CCN and INP, respectively, at cloud level. Based on this initial data set, we demonstrate the potential of the C×C approach for spaceborne ACI studies on two examples: the sensitivity of $N_d$ and $r_{eff}$ to changes in $n_{CCN}$ and the relation of the occurrence of ice-containing clouds to changes in $n_{INP}$.

For liquid clouds, we find a $\Delta \ln N_d / \Delta \ln n_{CCN}$ of 0.13 to 0.30 which is at the lower end of commonly inferred values of

0.3 to 0.8 (Bellouin et al., 2020). This is likely the combined result of applying $n_{CCN}$ at cloud level rather than column aerosol proxies and the fact that the considered isolated clouds are less susceptible to aerosol perturbations than cloud decks with larger cloud fraction. The inferred $\Delta \ln r_{eff} / \Delta \ln n_{CCN}$ between -0.09 and -0.21 means that $r_{eff}$ decreases by -0.81 to -3.78 nm per increase in $n_{CCN}$ of $1\,cm^{-3}$.

For ice-containing clouds, we find a tendency towards more cloud ice and more fully glaciated clouds with increasing $n_{INP}$

that cannot be explained by the increasingly lower $T_{top}$ of super-cooled liquid, mixed-phase, and fully glaciated clouds alone. This finding is in agreement with earlier studies in which $n_{INP}$ is approximated by the concentration of mineral dust from aerosol transport modelling and reanalysis (Seifert et al., 2010; Villanueva et al., 2020).

The purpose of this publication is to present the novel C×C approach as well as its first preliminary results for low-level liquid and mixed-phase clouds from application to one year of data over a comparably small section of the full Earth disk

covered by MSG-SEVIRI. A more comprehensive application of the C×C approach requires a significant increase in the volume of data that is available for detailed analysis including further stratification according to meteorological parameters to better confine cloud-controlling factors. In future work, the strong limitation in data yield imposed by the matching and quality-assurance constraints will be addressed by:

1. **Extending the considered time period** to exploit the full length of the time series of coinciding MSG and CALIPSO

satellite observations from 2006 to 2023. Given a comparable success rate as for 2014, this would provide a 17-fold increase of the volume of the matched aerosol-cloud data set for the region considered so far.

2. **Shifting or widening the study region** within MSG-SEVIRI's field of view. This allows for covering different cloud regimes, assessing the impact of different dominating aerosol types, and evaluating results with independent data from regional field experiments.

3. **Adapting the cloud screening method** to consider a larger set of cloud types such as cirrus and deep convection. These cloud types have rarely been considered in spaceborne studies and the C×C approach might provide a step forward in that respect.

4. **Expanding the application to other geostationary sensors** that cover further regions of the globe. This includes coverage of the Americas with GOES-R-ABI and of East Asia by the Advanced Himawari Imager (AHI, Bessho et al.

2016) aboard JAXA's Himawari-8. Both sensors also feature an improved spatial and temporal resolution compared to MSG-SEVIRI.



5. **Adapting the retrieval of cloud-relevant aerosol properties** to spaceborne lidar missions beyond CALIPSO, such as the Atmospheric Lidar on the Earth Cloud, Aerosol and Radiation Explorer (EarthCARE, Wehr et al. 2023) which is set for launch in 2024, to extend the time range that can be considered for ACI studies with the C×C approach.

6. **Refining and further developing the methods** for assessing different ACI mechanisms from the inferred data set of matched aerosol-cloud cases. The added time dimension related to resolving cloud development calls for dedicated and creative analysis approaches that best exploit the novel information.

For now, auxiliary information on meteorological parameters such as temperature, humidity, and pressure is taken from the CALIPSO Aerosol Profile product, as it is also needed for deriving CCN and INP concentrations. Future developments of our
methodology will include a more comprehensive consideration of meteorological information – specifically of confounding factors such as relative humidity, lower tropospheric stability, and vertical velocity – from meteorological reanalysis data.

Finally, continuity to the MSG-SEVIRI-CALIPSO data matching as well as advanced data products from next-generation sensors will soon be available in the form of combining geostationary observations with the Flexible Combined Imager (FCI) on Meteosat Third Generation (MTG-I1, Holmlund et al. 2021) which was launched in 2022 with the polar-orbiting lidar
observations provided by EarthCARE which is currently on schedule for launch in May 2024.



*Code availability.* The TrackMatcher code (Bräuer and Tesche, 2022) is available at https://github.com/LIM-AeroCloud/TrackMatcher.jl (last access: 23 November 2023).

*Data availability.* CALIPSO level 2 aerosol profiles can be accessed at https://doi.org/10.5067/CALIOP/CALIPSO/LID\T1\L25KMAPRO-STANDARD-V4-20 (last access: 23 November 2023). CLAAS-2 data are available at https://doi.org/10.5676/EUM/CLAAS/V002 (last access: 23 November 470 2023). The global gridded CCN data set (Choudhury and Tesche, 2023) is available at https://doi.pangaea.de/10.1594/PANGAEA.956215 (last access: 23 November 2023).

*Author contributions.* MT and PA conceived the presented approach. FA and MT developed the matching methodology. FA and GC developed the INP and CCN retrievals. FM and TS developed the cloud-tracking methodology. All authors contributed to the revision of the methodology, the discussion of the findings, and participated in writing the original manuscript.

*Competing interests.* The authors declare no competing interests.

*Acknowledgements.* This research has been supported by the Franco-German Fellowship Program on Climate, Energy, and Earth System Research (Make Our Planet Great Again-German Research Initiative (MOPGA-GRI), grant number 57429422) of the German Academic Exchange Service (DAAD), funded by the German Ministry of Education and Research. This research has also been supported by the Federal State of Saxony and the European Social Fund (ESF) in the framework of the program "Projects in the fields of higher education and 480 research" (grant no. 100649813).



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



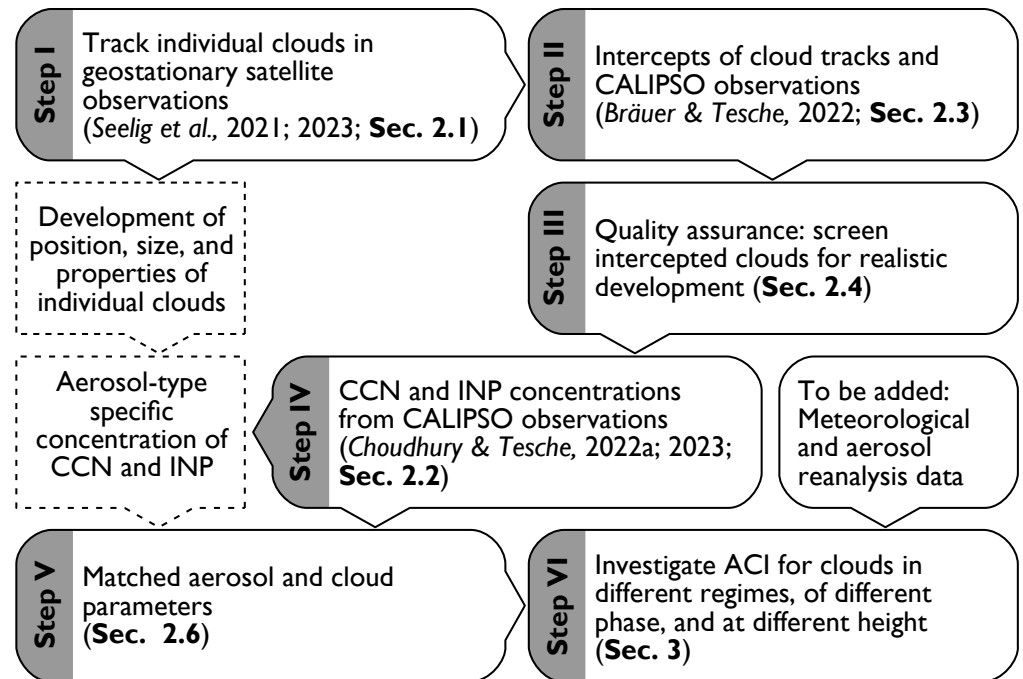

**Figure 1.** Flow chart of the different steps in the data analysis (solid boxes) with references to the corresponding literature and sections in this paper and the inferred data products (dashed boxes).

**Table 1.** Sensitivity of $r_{\mathrm{eff}}$ and $N_{\mathrm{d}}$ to changes in $n_{\mathrm{CCN}}$. Values refer to the full range of $n_{\mathrm{CCN}}$ represented by the three quantiles in Figure 12 (all) and the change from clean to moderately polluted conditions (lower two) and are based on median and mean values, respectively. The unit for $\Delta r_{\mathrm{eff}}/\Delta n_{\mathrm{CCN}}$ is nm/cm$^{-3}$ while all other parameters have no units. However, $\Delta N_{\mathrm{d}}/\Delta n_{\mathrm{CCN}}$ is multiplied by 100, i.e., refers to a $\Delta n_{\mathrm{CCN}}$ of 100 cm$^{-3}$.

| | based on medians | | based on means | |
|---|---|---|---|---|
| | all | lower two | all | lower two |
| $\Delta r_{\mathrm{eff}}/\Delta n_{\mathrm{CCN}}$ | -0.81 | -1.94 | -1.42 | -3.78 |
| $\Delta \ln r_{\mathrm{eff}}/\Delta \ln n_{\mathrm{CCN}}$ | -0.09 | -0.14 | -0.14 | -0.21 |
| $\Delta N_{\mathrm{d}}/\Delta n_{\mathrm{CCN}}$ | 1.17 | 2.30 | 1.51 | 1.48 |
| $\Delta \ln N_{\mathrm{d}}/\Delta \ln n_{\mathrm{CCN}}$ | 0.23 | 0.30 | 0.19 | 0.13 |




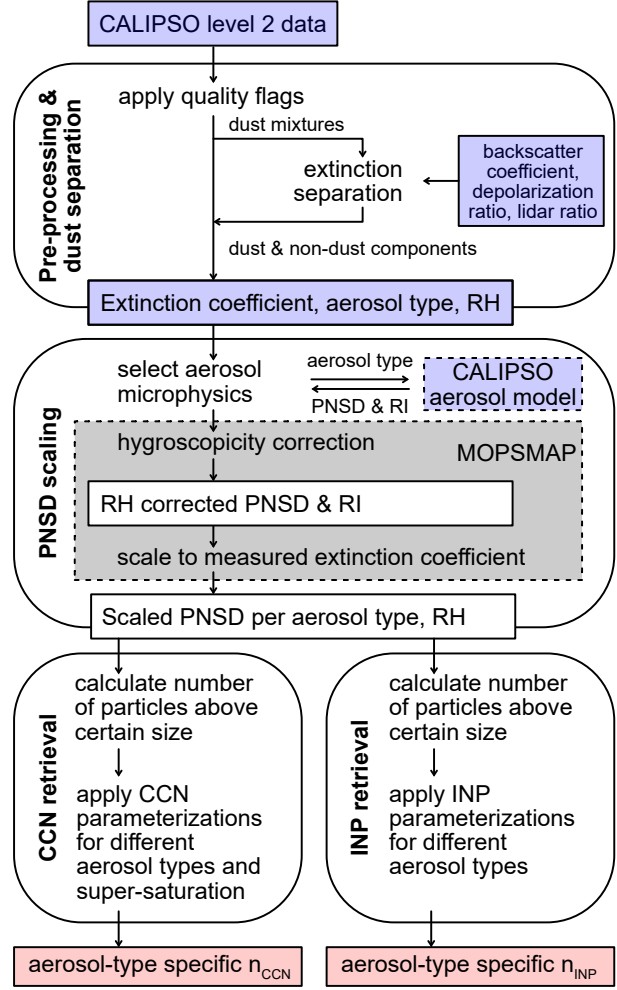

**Figure 2.** Flowchart of the OMCAM retrieval for inferring $n_{CCN}$ and $n_{INP}$ from spaceborne CALIPSO lidar observations. Input from CALIPSO measurements and the CALIPSO aerosol model is marked by blue boxes. The grey area covers the hygroscopicity correction and particle number size distribution (PNSD) scaling with the help of MOPSMAP. White boxes mark intermediate products. The final $n_{CCN}$ and $n_{INP}$ for different aerosol types are given in the red boxes. RH and RI relate to relative humidity and refractive index, respectively. Figure revised from (Choudhury and Tesche, 2022a).





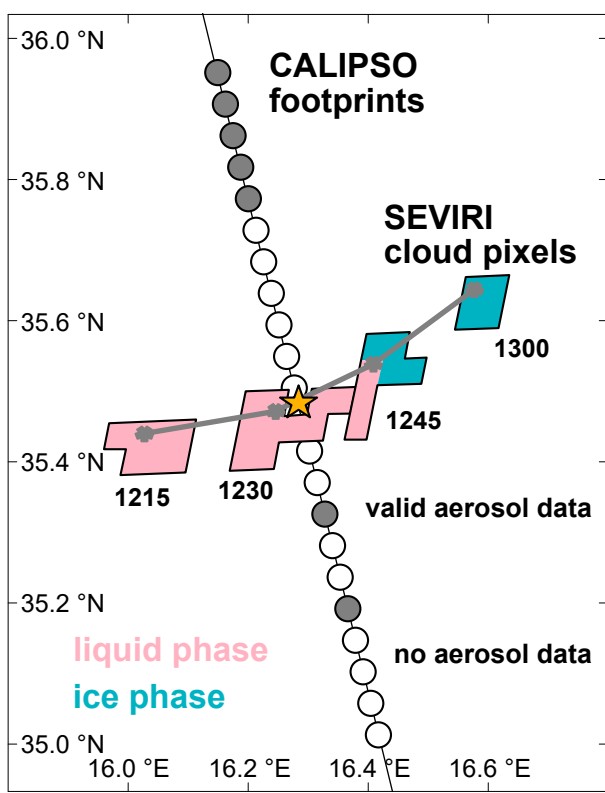

**Figure 3.** Example for matching a SEVIRI cloud trajectory from 1215 to 1300 on 20 March 2014 (grey line connecting the centroids of each time step) with the 5-km averaged footprints along the CALIPSO ground track for an overpass at 1215 on 20 March 2014. Filled CALIPSO footprints mark profiles with valid aerosol data while empty circles refer to profiles without aerosol information. The yellow star marks the intercept point. Light red and blue areas refer to SEVIRI pixels for which cloud water is identified as liquid and ice, respectively.



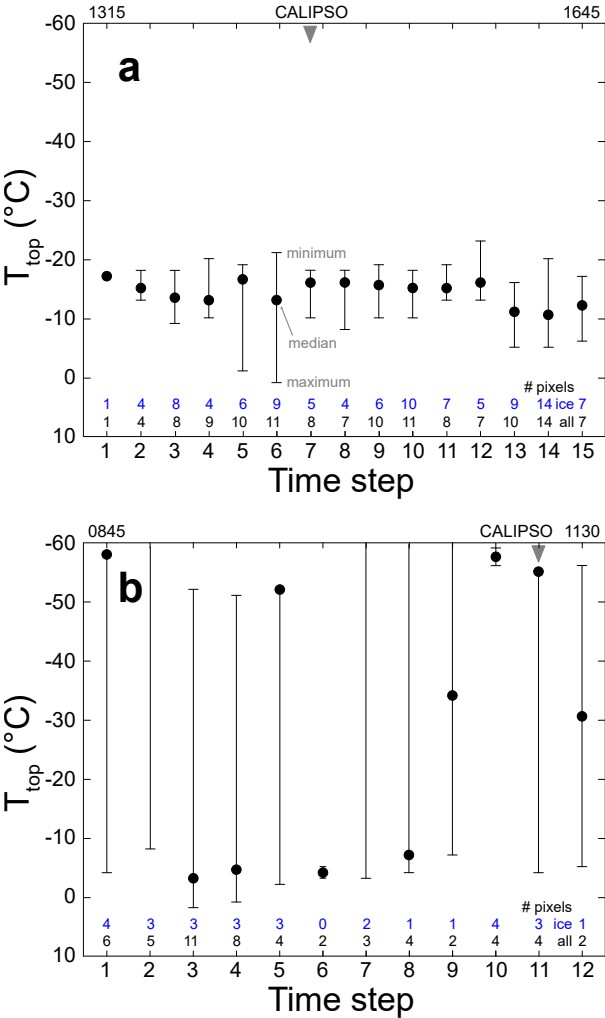

**Figure 4.** Realistic (a) and unrealistic (b) development of $T_{top}$, cloud area, and cloud phase found on 14 August 2014 and 24 March 2014, respectively. $T_{top}$ per time step refers to the median over all cloud pixels (black dot) while the range marks the temperatures of the warmest (lower whisker) and coldest (upper whisker) pixel, respectively. Numbers at the bottom refer to the size of the cloud in pixels (black) and the number of pixels identified as ice (blue). The start and end times (in UTC) of the trajectories and the time of the CALIPSO observation (grey triangle) are given at the top of the plot.



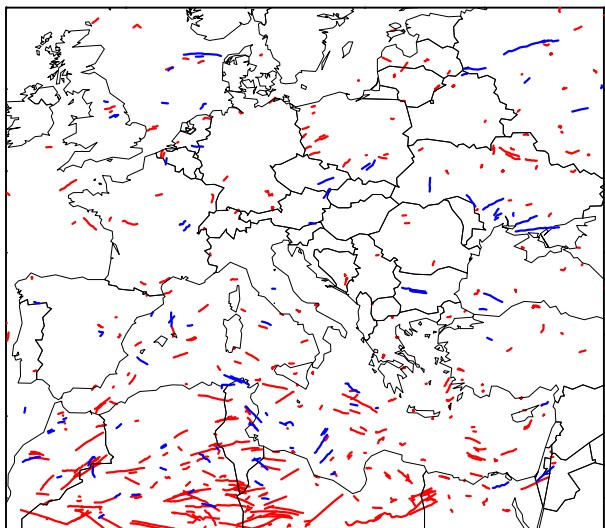

**Figure 5.** Overview of the study region with trajectories of liquid-only (red, $N = 399$) and ice-containing (blue, $N = 95$) clouds that could be matched to surrounding $n_{CCN}$ and $n_{INP}$, respectively.

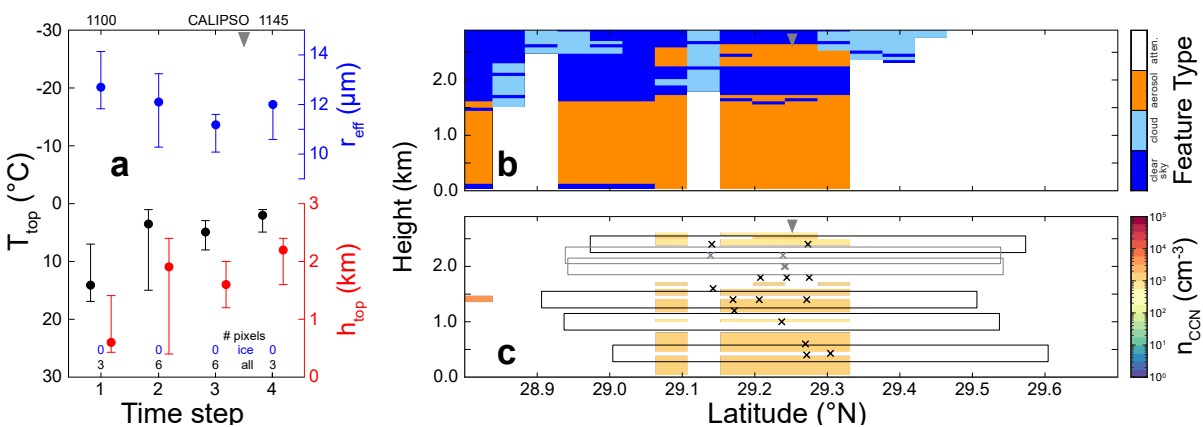

**Figure 6.** Example of (a) the development of $T_{top}$ (black), $h_{top}$ (red), and $r_{eff}$ (blue) for a liquid-only cloud observed by MSG-SEVIRI between 1100 and 1145 UTC on 18 March 2014 and the surrounding aerosol field from CALIOP observations in terms of (b) the feature type and (c) $n_{CCN}$ for an overpass at 1137 UTC. The grey triangles mark the intercept between the cloud trajectory and the CALIPSO ground track. The example boxes in (c) illustrate the along-track distance and height range centred around the location and $h_{top}$ (crosses) of individual pixels in (a) used for matching aerosol and cloud data. Successful matches, i.e. valid CCN data is available within a set distance to a pixel of a tracked cloud, are given in black while matches without valid CCN data are marked grey. White in (b) marks regions where the lidar signal has already been fully attenuated from clouds above. Note that not all detected aerosol features (orange in b) provide CCN information as the result of applying quality flags in the CCN retrieval (see Figure 2).



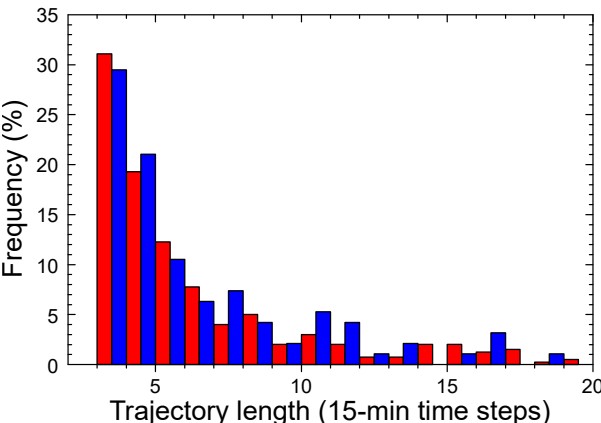

**Figure 7.** Frequency of occurrence of trajectories of different length in 15-min time steps for trajectories of 399 tracked liquid-only (red) and 95 ice-containing clouds (blue) that could be matched to CALIPSO overpasses with valid aerosol data.

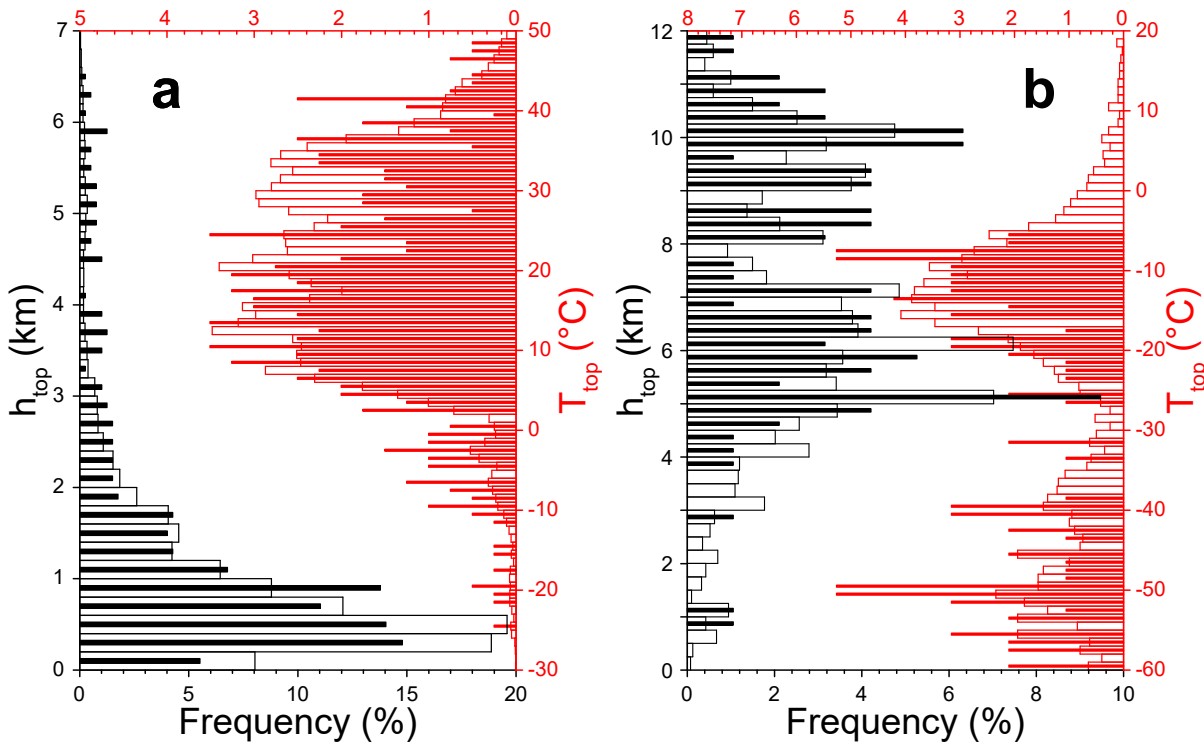

**Figure 8.** Frequency of occurrence of $h_{top}$ (black) and $T_{top}$ (red) of liquid (a) and ice-containing clouds (b) that could be matched to CALIPSO overpasses with valid aerosol data. Open bars refer to pixel-wise information (35799 for liquid-only, 4016 for ice-containing) and solid bars mark whole-trajectory medians (399 for liquid-only, 95 for ice-containing). Note that data relating to the left and right scales, respectively, are independent of each other.



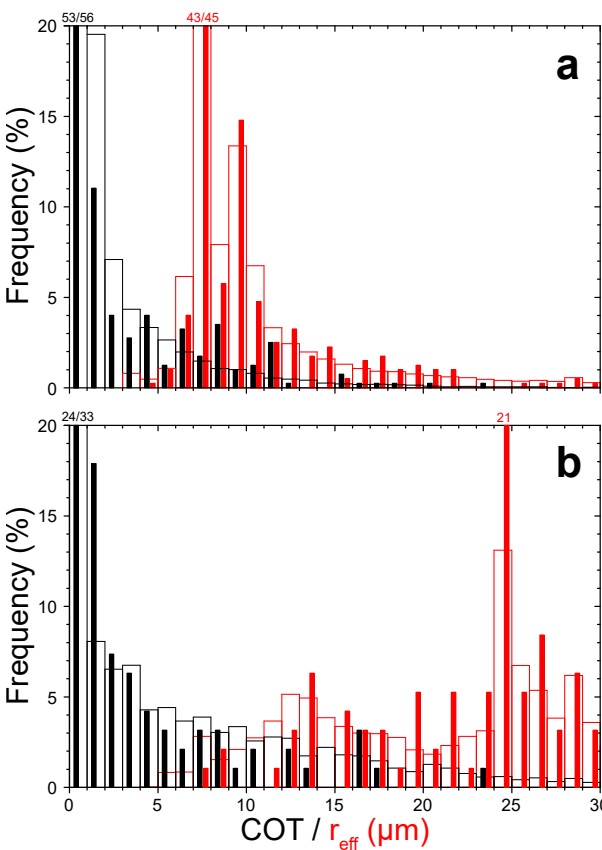

**Figure 9.** Frequency of occurrence of COT (black) and $r_{eff}$ (red) for (a) liquid-only and (b) ice-containing clouds that could be matched to CALIPSO overpasses with valid aerosol data. Note that the number of pixels with valid cloud information is reduced compared to Figure 8. Open bars refer to pixel-wise information (24089 out of 35799 available pixels for liquid-only clouds and 3769 out of 4016 available pixels for ice-containing clouds) and solid bars mark whole-trajectory medians (399 for liquid-only, 95 for ice-containing). Numbers at the top of the plots refer to the pixel-wise/whole-trajectory values related to the intervals for which the frequency of occurrence exceeds the scale.



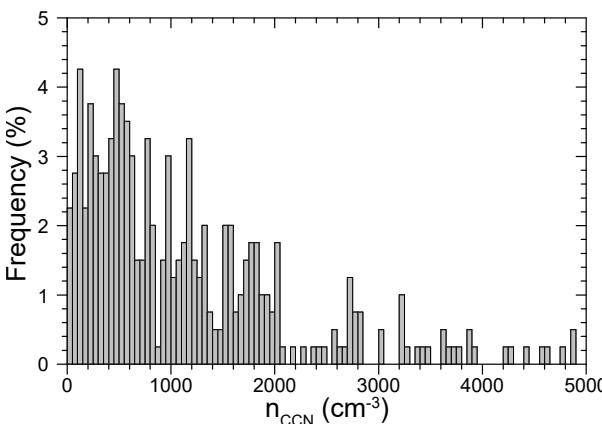

**Figure 10.** Frequency of occurrence of median $n_{CCN}$ for 399 CALIPSO overpasses that could be matched to tracked liquid-only clouds.

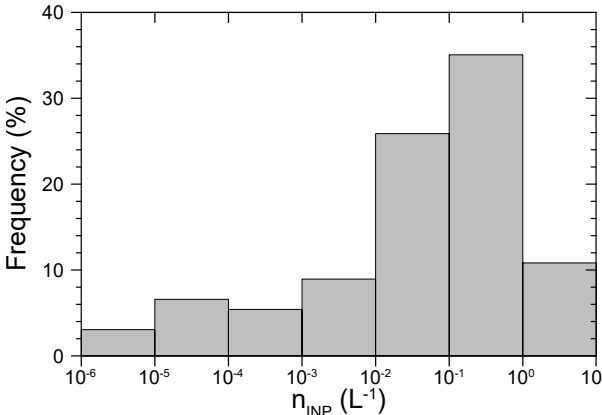

**Figure 11.** Frequency of occurrence of median $n_{INP}$ for 95 CALIPSO overpasses that could be matched to tracked ice-containing clouds.



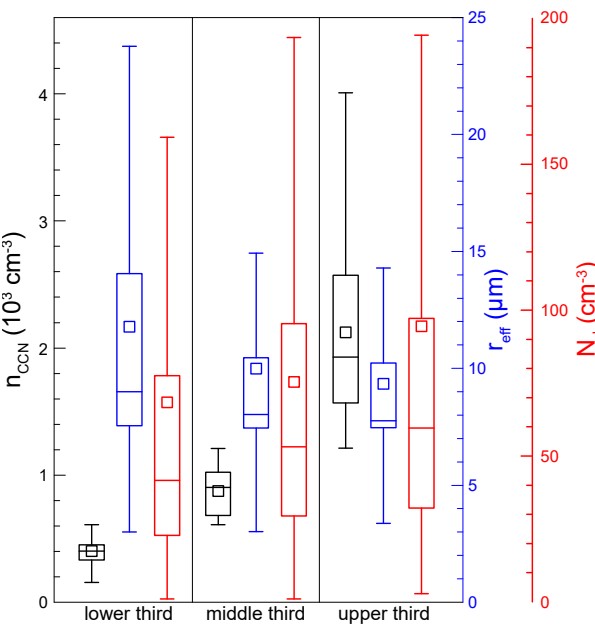

**Figure 12.** Boxplots of $n_{CCN}$ (black), $r_{eff}$ (blue), and $N_d$ (red) for 16230 matches of cloud pixels and aerosol information sorted into three quantiles. See text for details.

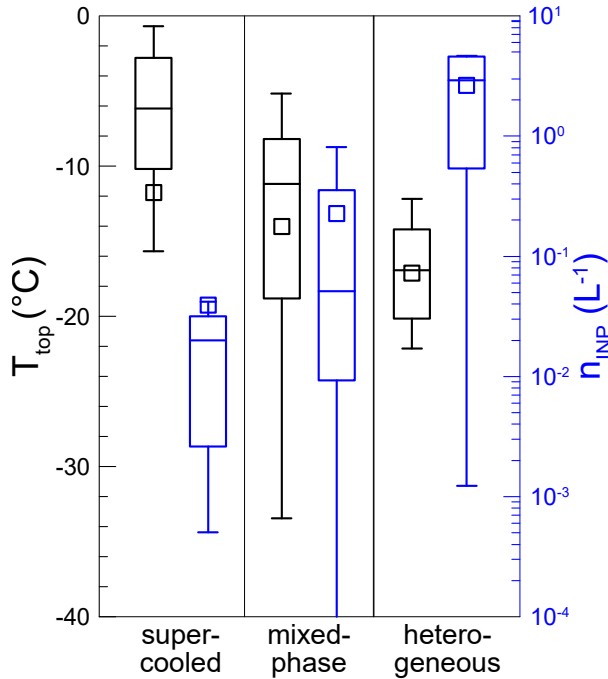

**Figure 13.** Boxplots of cloud-median $T_{top}$ (black) and $n_{INP}$ (blue) for super-cooled liquid clouds (27 cases), mixed-phase clouds (45 cases), and heterogeneously frozen ice clouds (8 cases) as determined with the C×C approach. See text for details.