# Peer review of "A cloud-by-cloud approach for studying aerosol-cloud interaction in satellite observations"

_EGUsphere, 2023_

## Author Comment (AC1)

We would like to thank the two referees for dedicating their time to improve the quality of our manuscript. Please find our point-by-point reply (blue) to all referee comments (black) below:

**Anonymous Referee #1**

The presented paper constitutes a significant contribution to the field of atmospheric science and aerosol-cloud interactions. The authors introduce a Cloud-by-Cloud (C×C) approach, merging geostationary satellite observations with polar-orbiting lidar data, to systematically study aerosol-cloud interactions. By matching cloud development with height-resolved Cloud Condensation Nuclei (CCN) and Ice Nucleating Particle (INP) concentrations, they investigate the aerosol cloud interactions. The application of this novel methodology to data from MSG-SEVIRI and CALIPSO over Europe and Northern Africa in 2014 yields compelling insights into the sensitivity of liquid and ice-containing clouds to changes in aerosol concentrations. This methodological innovation provides a platform for advancing our understanding of atmospheric dynamics and climate processes.

The overall study is well-written, well-structured, characterized by scientific sufficiency and therefore falls into the scope of EGU community, thus could be published as paper. I only have some minor revisions/ comments below.

We thank the referee for the very positive evaluation of our work.

In Figure 1, the authors present a flow chart outlining the various steps in the data analysis. This visual aid serves as a guide for readers, providing references to the relevant literature and corresponding sections in the paper, enhancing the accessibility of the complex data analysis process. This is a helpful asset in understanding the study's methodology in the first look.

We are happy to hear that Figure 1 worked as intended.

In Subsection 2.1 maybe consider adding a brief explanation or reference for Particle Image Velocimetry (Adrian and Westerweel, 2010) for readers who may not be familiar with the technique.

We have added to following statement to clarify that Particle Image Velocimetry is based on the cross-correlation of identified objects in consecutive images:
"*This process determines the displacement that best matches objects in a pair of consecutive binary 2d images by means of cross-correlation.*"

The criteria for filtering the cloud trajectories based on cloud top height, clear air development, and daytime occurrence are well-defined and reasonable. It might be helpful to mention the concept behind choosing a factor of 4 for the difference in area threshold.

We agree that a justification of this choice is warranted. The purpose of the factor is to exclude clouds with unrealistic growth or decay rates which applies particularly at the lower end of the cloud-size spectrum. While large clouds might double or triple in size (or decay by the same factors) over two consecutive time steps, it is our experience that an aerial change of a factor of four is very rare. However, the data set also includes a large number of small clouds that might only consist of a few pixels and could show large size changes when

involved in merging or splitting event. We hence chose a factor of four as aerial threshold to ensure realistic cloud development for both small and large clouds. The following statement has been added to the text:

*"This areal threshold ensures realistic growth (decay) rates for non-merging (non-splitting) clouds and is based on our observations of small clouds – particularly the growth of one-pixel and two-pixel clouds."*

In subsection 2.2, the flow chart of the OMCAM algorithm is a useful addition, aiding in the comprehension of the complex aerosol concentration retrieval process. The expansion of OMCAM to derive INP concentrations is an important extension. Clearly articulating this expansion and its relevance enhances the understanding of the methodology.

We are not certain if this is a statement or a suggestion for improvement. Assuming the latter, we have revised the final paragraph of Subsection 2.2 to:

*"For this work, OMCAM was expanded analogous to Mamouri and Ansmann (2016) and Marinou et al. (2019) to estimate the concentration of INP reservoir particles which are subsequently used as input to INP parametrizations to obtain $n_{INP}$ (lower right panel in Figure 2). This capability allows for extending the scope of the C×C approach towards studying aerosol effects on ice-containing clouds. In contrast to $n_{CCN}$, independent in-situ measurements of $n_{INP}$ are sparse. This is why the validation of OMCAM-derived values is still a matter of ongoing research. Nevertheless, initial findings based on comparison to airborne in-situ measurements indicate that CALIPSO-derived $n_{INP}$ have a quality comparable to those inferred from ground-based lidar measurements (Marinou et al., 2019; Choudhury et al., 2022)."*

Reference to Figure 2: Consider briefly summarizing the key steps in the OMCAM algorithm or referring to specific elements in Figure 2 to direct the reader's attention to the relevant parts.

We have added references to the corresponding panels in the OMCAM flow-chart in Figure 2 to the description of the OMCAM algorithm in Subsection 2.2.

In subsection 2.3 the introduction of TrackMatcher and its purpose is clear. Consider briefly mentioning the significance of intercept points between cloud trajectories and the CALIPSO ground track in the context of the overall research objective. The step-by-step breakdown of the algorithm is well-detailed, helping the readers who are not familiar with the TrackMatcher tool. A brief context for the operator-set auxiliary information extracted at or around the intercept points would also be helpful.

Adding specifics on the significance of TrackMatcher for the applicability of the overall approach is an excellent suggestion. We have added the following statements:

*"In the context of the C×C approach, this enables to match cloud properties along a cloud trajectory from geostationary satellite observations (Section 2.1) to the surrounding cloud-relevant aerosol concentrations from polar-orbiting lidar observations (Section 2.2)."*

*"For applying the C×C approach, the auxiliary formation extracted from the CALIPSO level 2 aerosol profile product consists of height profiles of the OMCAM input parameters, i.e., backscatter and extinction coefficients, particle linear depolarization ratio, vertical feature mask, and cloud-aerosol-discrimination (CAD) score."*

The explanation regarding the time difference between cloud tracks and CALIPSO satellite is clear. However, it might be helpful to explain why the chosen range of ±90 minutes was suitable for this study.

We would like to clarify that the range of ±90 minutes time delay between the observations is the result of applying the retrieval without a constraint on time difference rather than pre-set chosen value. We believe that the time range is suitable for this study as aerosol fields generally don't vary too much over such a time period. We have added the following statement for clarification:
"*This time difference is well below the time scale for which major changes in an aerosol field could be expected (Anderson et al., 2003; Kovacs, 2006).*"

Anderson, T. L., Charlson, R. J., Winker, D. M., Ogren, J. A., and Holmén, K.: Mesoscale Variations of Tropospheric Aerosols, J. Atmos. Sci., 60, 119-136, https://doi.org/10.1175/1520-0469(2003)060<0119:MVOTA>2.0.CO;2, 2003.

Kovacs, T.: Comparing MODIS and AERONET aerosol optical depth at varying separation distances to assess ground-based validation strategies for spaceborne lidar, J. Geophys. Res., 111, D24203, https://doi.org/10.1029/10.1029/2006JD007349, 2006.

Providing a brief clarification on what criteria determine the dismissal of CALIPSO aerosol data would enhance understanding.

We have added a statement that clarifies that we are following the criteria summarized in Table 1 of *Tackett et al.* (2018).

Tackett, J. L., Winker, D. M., Getzewich, B. J., Vaughan, M. A., Young, S. A., and Kar, J.: CALIPSO lidar level 3 aerosol profile product: version 3 algorithm design, Atmos. Meas. Tech., 11, 4129-4152, https://doi.org/10.5194/amt-11-4129-2018, 2018.

The addition of Figure 3 is particularly beneficial for visualizing the described process.

Thank you.

Subsection 2.4 describes the quality-assurance process. The four criteria for assessing realistic cloud development are well-detailed. Lines 214-216: Consider rephrasing to avoid redundancy, e.g., "if more than one of the criteria is met" could be streamlined to "if multiple criteria are met." Also, the authors could articulate the context behind the high level of scrutiny and data reduction and explain how this rigorous approach ensures the physical meaningfulness of findings from the bottom-up database.

We have rephrased as suggested. We have also added the following text to highlight that only physically meaningful input can provide physically meaningful output:

"*While this rigorous screening of tracked clouds implies a high level of data reduction, it ensures that only cases in which the observations show a cloud that develops in a physically meaningful way along its trajectory, i.e., that are not affected by any of the limitations of passive remote sensing, will be considered in the subsequent investigation of aerosol-cloud interactions. In other words, we can only obtain robust findings from the bottom-up data*

*base if the highest level of scrutiny is applied in the identification trustworthy aerosol-cloud cases.*"

In subsection 3.2 the rationale for excluding the upper and lower 5% of cases is clear. Consider briefly mentioning the impact of this exclusion on data analysis.

We have added the following statement to account for the Referee's comment:
"*While the exclusion of unrealistic values of $n_{CCN}$ doesn't affect the data set presented here, it will become more crucial as the volume of the bottom-up data set increases.*"

The comparison with studies based on passive CCN proxies adds context (Subsection 3.3). However, consider expanding on the potential reasons for the differences observed, providing a more nuanced interpretation. The mention of future research and the need for a longer time series is appropriate.

Thank you for the suggestion. We have revised the text as follows to provide further context regarding the lower sensitivity:
"*The latter values are at the lower end of earlier findings of 0.3 to 0.8 based on column CCN proxies (Bellouin et al., 2020). There are likely two factors at play. First, our study is the first one to derive sensitivities based on inferred CCN concentrations at cloud level. On the one hand, this ensures that we only consider cases in which the observed aerosols can actually interact with the observed clouds. On the other hand, cloud-layer $n_{CCN}$ is likely to give a weaker aerosol signal than a column proxy, potentially reducing the deduced sensitivity. Second, the isolated clouds considered in the current data set might not be as sensitive to changes in aerosol concentration as, e.g., widespread stratocumulus clouds (Bellouin et al., 2020; Gryspeerdt et al., 2017; Goren et al., 2019) or their lifetime might be too short for revealing the full impact of the aerosol perturbation (Glassmeier et al., 2021; Gryspeerdt et al., 2021). Future research based on a larger C×C data set will reveal if the lower sensitivity can be corroborated and if the regional variation of sensitivity of cloud parameters to changes in $n_{CCN}$ can also be resolved with our new approach.*"

Glassmeier, F., Hoffmann, F., Johnson, J. S., Yamaguchi, T., Carslaw, K. S., and Feingold, G.: Aerosol-cloud-climate cooling overestimated by ship-track data, Science, 371, 485-489, https://doi.org/10.1126/science.abd3980, 2021.

The discussion of the temperature-dependent relationship between T_top and n_INP is insightful (Subsection 3.4). Consider emphasizing the implications of this finding for a more nuanced understanding of aerosol-cloud interactions. The comparison to studies using dust concentrations as an INP proxy is also valuable. Briefly discussing potential differences and similarities in findings could be helpful. The emphasis on the need for a larger data set for more robust conclusions is well-stated.

We have added the following statement to the corresponding paragraph to better resolve the differences to the analysis based on dust concentration as INP proxy:

"*Those studies address the issue by resolving the occurrence of ice-containing clouds within different ranges of $T_{top}$ for different ranges of background aerosol concentrations while our approach directly relates $n_{INP}$ at cloud level to the occurrence of ice in the matched clouds. However, resolving a consistent picture through an entirely different data analysis concept*

*increases our overall confidence in applying the C×C to both liquid-water and ice-containing clouds.*"

The Summary and outlook section is well-organized and effectively communicates the achievements of the study, as well as the planned future directions. Enhancing the interpretative aspects and providing a bit more context for readers less familiar with the field could further strengthen the summary. Here are some questions that could possibly be answered in this section:

What could the observed sensitivities mean for our understanding of aerosol-cloud interactions on a broader scale?

While there is a mention of the findings being at the lower end of commonly inferred values, consider further discussing the significance of this difference. How might this influence the broader understanding of aerosol-cloud interactions, and what factors could contribute to these variations?

In the outlook section, provide a bit more detail on the rationale behind each future step. For instance, why is shifting or widening the study region important, and how might it enhance the robustness of the study?

Briefly discuss the potential impacts or applications of the research, especially if successful. How might the findings contribute to our understanding of climate processes or inform future satellite missions?

Thank you for the suggestions. We have addressed them by adding a discussion of the factors contributing to lower sensitivities to Section 3.3 as stated above. We have revised the list of future actions to account for the raised items. We have also added the following statement to the summary paragraph on the sensitivities to highlight the implication for climate studies: *"While further work is needed to corroborate the robustness of these findings, they would imply a weaker cooling related to aerosol-cloud interactions than currently accounted for in studies of anthropogenic climate change."*

**Anonymous Referee #2**

The paper by Alexandri et al. introduces a Cloud-by-Cloud (C×C) approach, merging geostationary satellite observations with polar-orbiting lidar data, to assess aerosol-cloud interactions and demonstrates the application of the C×C approach on some studies. In general, the authors provide the necessary information and guidance to understand the C×C approach and the manuscript is well written and structured. I recommend the manuscript for publication after some comments detailed below in my review.

We thank the referee for the positive evaluation of our work. We have addressed the comments as outlined in the point-by-point reply below.

**Minor comments:**

Line 125. "The best match within set thresholds is kept if the difference in area does not exceed a factor of four." The authors must explain the reason for choosing a factor of 4 in the cloud-tracking analysis. Is it an empirical estimate?

We agree that a justification of this choice is warranted. The purpose of the factor is to exclude clouds with unrealistic growth or decay rates which applies particularly at the lower end of the cloud-size spectrum. While, large clouds are unlikely to double or triple in size over two consecutive time steps, the data set also includes a large number of small clouds that might only consist of a few pixels. In our data, we found that non-merging one-pixel and two-pixel clouds generally don't quadruple in size and, hence, chose a factor of 4 as areal threshold. The following statement has been added to the text:
"*This areal threshold ensures realistic growth (decay) rates for non-merging (non-splitting) clouds and is based on our observations of small clouds – particularly the growth of one-pixel and two-pixel clouds.*"

Line 178-179. "In this study, we didn't limit the time difference for finding matches between a cloud track and the CALIPSO satellite but most cases fell within a range between 0 and ±90 minutes." The time difference can be a critical factor in the retrievals. Why didn't authors try to limit the criteria? How different could be the results?

Time difference is parameter that is provided in the TrackMatcher output and larger data sets could certainly be used to assess the impact of time difference by stratifying the results according to different intervals of time difference. We believe that the covered time range is suitable for demonstrating the feasibility of the C×C approach as aerosol fields generally don't vary too much over such a time period. We have added the following statement for clarification:
"*This time difference is well below the time scale for which major changes in an aerosol field could be expected (Anderson et al., 2003; Kovacs, 2006).*"

Anderson, T. L., Charlson, R. J., Winker, D. M., Ogren, J. A., and Holmén, K.: Mesoscale Variations of Tropospheric Aerosols, J. Atmos. Sci., 60, 119-136, https://doi.org/10.1175/1520-0469(2003)060<0119:MVOTA>2.0.CO;2, 2003.

Kovacs, T.: Comparing MODIS and AERONET aerosol optical depth at varying separation distances to assess ground-based validation strategies for spaceborne lidar, J. Geophys. Res., 111, D24203, https://doi.org/10.1029/10.1029/2006JD007349, 2006.

Consider moving Figure 4 to appendix.

We prefer to retain Figure 4 in the text as it provides valuable context regarding the need to ensure that considered clouds develop realistically.

Figure 6 should be made larger, as it is not readable in the current size.

Thank you for the comment. We have adapted the width of panels b and c to increase the overall size of the figure.

Figure 8b. Should explain more the existence of the second peak of the ice-containing clouds group and also, provide a reference for their explanation.

Thank you for the suggestion. We have revised the text to:

"*The other group with $T_{top} > -38$ °C covers the regime of heterogeneous freezing, i.e., INP are needed to form cloud ice (Kanji et al., 2017), and represents the temperature range at which supercooled, mixed-phase, and ice clouds can be observed. We expect that the latter clouds would be most susceptible to changes in aerosol concentrations. They make up 2673 pixels (53 clouds) with $h_{top}$ ranging from 0.4 to 9.7 km (0.8 to 9.2 km) and a median of 5.8 km (5.7 km).*"

Line 374-376. It would be better to stress earlier the limitations of the C×C approach.

The mentioned lines don't really refer to a limitation of the C×C approach but to the limited abundance of CCN data related to aerosol types other than mineral dust and polluted continental in the data set inferred from the initial application of the new presented here. The statement is at the appropriate position in the text as it refers to an outcome related to the creation of the matched aerosol-cloud data set and clarifies why we are currently not able of further refining the investigation of CCN effects on liquid water clouds according to different aerosol types. We are certain that this lack of diversity in considered aerosol types will be overcome by our future activities as outlined in the list of actions in Section 4.